# Estimation of Respiratory Syncytial Virus-attributable hospitalizations among older adults in Japan between 2015 and 2018: An administrative health claims database analysis

Masafumi Seki[1], Yasuhiro Kobayashi[2], Estelle Méroc[3], Takahiro Kitano[2], Aleksandra Polkowska-Kramek[3], Asuka Yoshida[2], Shuhei Ito[2], Caihua Liang[4]*, Robin Bruyndonckx[3], Solomon Molalign Moges[3], Eduardo Conde-Sousa[3], Charles Nuttens[5], Bradford D. Gessner[4], Elizabeth Begier[6]

1 Division of Infectious Diseases and Infection Control, Saitama Medical University International Medical Center, Hidaka, Japan, 2 Pfizer Japan Inc., Tokyo, Japan, 3 P95 Clinical and Epidemiology Services, Leuven, Belgium, 4 Pfizer Inc., New York, United States of America, 5 Pfizer Inc., Paris, France, 6 Pfizer Inc., Dublin, Ireland

* caihua.liang@pfizer.com

## Abstract

We estimated the incidence rates (IRs) of respiratory syncytial virus (RSV)-related hospitalizations among adults aged ≥60 years in Japan from 2015 to 2018, using a time-series model-based approach to better understand the disease burden. We obtained hospitalization data from the Medical Data Vision (MDV) database, restricted to Diagnosis Procedure Combination (DPC) hospitals. We estimated the annual age-specific RSV-attributable IR of hospitalizations for five cardiorespiratory outcomes based on selected ICD-10 codes employing a quasi-Poisson regression model. We projected our results to all DPC hospitals in Japan and also to all Japanese hospitals. In adults aged ≥60 years, the annual IR of RSV-attributable cardiorespiratory hospitalizations at DPC hospitals was estimated at between 100 and 124 per 100,000 person-years, projecting to 134–229 cardiorespiratory hospitalizations per 100,000 person-years at all hospitals. For respiratory hospitalizations at DPC hospitals, the annual IR was from 69 to 85 per 100,000 person-years (projecting to 96–157 hospitalizations per 100,000 person-years at all hospitals). IRs for all outcomes were consistently higher among adults aged ≥80 years than those 60–79 years. Our results indicate a high burden of RSV-attributable hospitalizations in older adults in Japan, highlighting the need to implement effective RSV prevention strategies.

**Data availability statement:** The datasets generated during and/or analyzed during the current study are not publicly available. The data are owned by a third-party, Medical Data Vision (MDV) Co., Ltd, and the authors do not have permission to share the data. The data are available upon request to MDV (https://en.mdv.co.jp; contacts: Hiroyuki Tada [email: tada@mdv.co.jp] and Yuki Santo [email: santo_yuki@mdv.co.jp]). The statistical codes are available upon request to the following contacts: statistics department, P95 (email: stat@p-95.com), and Hannah Volkman (Pfizer Inc; email: Hannah.Volkman@pfizer.com).

**Funding:** This study was sponsored by Pfizer Inc. Several authors [EM, SMM, ECS, RB, and APK] are employees of P95, which received funding from Pfizer Inc. for the conduct of this study, and for the development and editorial support of this manuscript. The funder provided support in the form of salaries for some authors [YK, TK, AY, SI, CL, CN, BDG, and EB], but non-authors from Pfizer did not have any additional role in the study design, data collection and analysis, decision to publish, or preparation of the manuscript.

**Competing interests:** P95 received funding from Pfizer Inc. for the conduct of this study, and for the development and editorial support of this manuscript. Caihua Liang, Yasuhiro Kobayashi, Takahiro Kitano, Asuka Yoshida, Shuhei Ito, Bradford D. Gessner, Charles Nuttens, and Elizabeth Begier are employees of Pfizer and may own Pfizer stock. Estelle Méroc, Solomon Molalign Moges, Eduardo Conde-Sousa, Robin Bruyndonckx, and Aleksandra Polkowska-Kramek are employees of P95. This does not alter the authors' adherence to PLOS ONE policies on sharing data and materials.

## Introduction

Respiratory syncytial virus (RSV) is a common cause of respiratory tract infections in both children and adults [1]. Among adults, RSV disease encompasses several clinical presentations, ranging from mild symptoms to severe lower respiratory tract infection and exacerbation of chronic respiratory and cardiac disease. Annual RSV epidemics occur seasonally, primarily in colder months in temperate climates [2].

It was estimated that 787,000 (460,000–1,347,000) RSV-associated hospitalizations occurred among older adults in high-income countries in 2019 [3]. Morbidity and mortality in older adults with underlying medical conditions are high, including those with chronic lung and heart disease, immunocompromised patients, residents of long-term care facilities, as well as others [4].

The incidence of RSV disease among adults is challenging to measure since RSV disease does not have a specific symptomatology that could distinguish cases from those of influenza and other respiratory viruses. Moreover, infrequent standard-of-care testing, lower diagnostic test sensitivity compared to children due to lower viral loads in respiratory secretions, low diagnostic capacity, and the high cost of polymerase chain reaction (PCR) testing all contribute to the underascertainment of RSV cases [5–7].

Due to considerable underascertainment and underreporting of RSV cases in adults, alternative model-based approaches have been developed to retrospectively estimate RSV infection incidence. These approaches involve analysis of data from administrative claims/electronic medical records and examining the association between changes in the viral activity indicator (e.g., virus-specific hospitalizations) and the corresponding changes in the incidence of the all-cause health outcome related to RSV, such as pneumonia, chronic obstructive pulmonary disease (COPD) exacerbation, or cardiorespiratory hospitalizations [8–10].

In Japan, the burden of RSV has been mostly studied in children [11]. In a multicentre prospective study in persons aged 15 years and older conducted from 2011 to 2013, RSV was reported to be the third most common cause of community-acquired pneumonia [12]. Additionally, a recent retrospective study conducted in Japan concluded that the average cost of hospital stays in RSV-diagnosed cohorts was much higher for older adults than for infants [13]. However, currently, there is no routine surveillance system in place for adults, and the clinical and economic burden of RSV in adults constitutes an important knowledge gap. A better understanding of RSV epidemiology in adults is critical since the population distribution of Japan is characterized by a large proportion of older adults, which has been projected to increase further in the coming decades, representing 41% in 2019 [14]. Further new RSV vaccines have been licensed, such as the protein subunit vaccines and mRNA vaccine, but assessing their utility at a population level depends on accurate burden estimates [15].

The aim of our study was to estimate the RSV-attributable hospitalization incidence among older adults in Japan. The study was based on a generic study protocol that was already implemented in several other countries [8,16,17].

## Methods

### Study design

We conducted a retrospective analysis to estimate the incidence rate (IR) of RSV-attributable hospitalizations. As described in Fig 1, this involved a statistical modeling approach that links the variation of RSV activity to that of all-cause cardiorespiratory hospitalizations while adjusting for influenza activity, seasonality, and underlying annual trends.

We obtained data from the Medical Data Vision (MDV) hospital-based database (data accessed on 10 May 2023), which includes individual patient data from administrative claims, Diagnosis Procedure Combination (DPC) data from hospitalizations, and outpatient visits of patients who attend hospitals participating in the DPC Per-Diem Payment System, which provide acute inpatient care. In the MDV database, data has been available since 2008, and spans over 485 hospitals and about 40 million patients as of August 2023. We included individuals 60 years and older hospitalized in Japan with one of the study outcomes in a hospital registered within the MDV database. Data were captured from 1 January

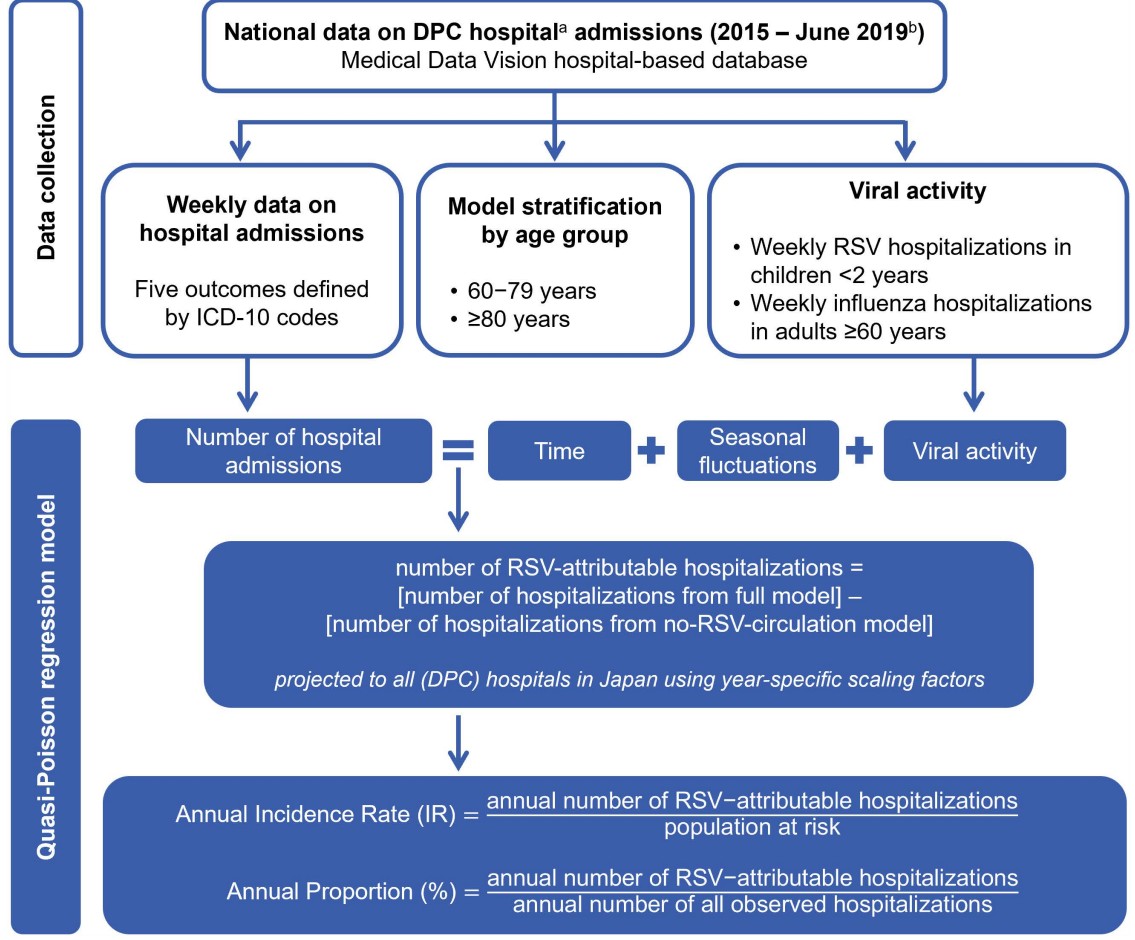

**Fig 1. Data collection and statistical analysis of the RSV model-based study conducted in Japan, 2015–2018.** *a*: Hospitals participating in the Diagnosis Procedure Combination system, which are mostly acute care hospitals; *b*: Data were captured from 1 January 2015 until 30 June 2019, to exclude the impact of the COVID-19 pandemic on RSV epidemiology; *Abbreviations*: DPC, Diagnosis Procedure Combination; ICD-10, International Classification of Diseases, 10th Revision; RSV, Respiratory Syncytial Virus.

2015 until 30 June 2019, to exclude the impact of the pandemic of COVID-19 on RSV epidemiology from the analysis. The data was received in an anonymized structured format, which ensured that individual patients could not be identified during or following the data collection process.

## Definition of outcomes and viral proxies

In the MDV source database, disease names and their ICD codes (10th revision) during hospitalization are listed for several items based on the DPC classification [18]. We defined outcomes by the occurrence of selected ICD-10 codes in disease name, which inputs the most medical resources, main disease name, and disease name behind hospitalization. By selecting these categories, we aimed to capture most accurately both primary and secondary diagnoses [19,20] and to exclude the categories that did not present clear seasonal patterns. As described in our generic protocol [8], we have included the selected cardiovascular disease codes most likely to be associated with RSV. We estimated the incidence of RSV-attributable hospitalizations for two main disease outcomes: cardiorespiratory diseases (J00–J99, I21, I48, I49, I50, I63, I64) and respiratory diseases (J00–J99). Additionally, we investigated three secondary outcomes, defined by subcategories of cardiorespiratory conditions that were found to be associated with RSV and presented clear seasonal patterns: influenza or pneumonia, chronic lower respiratory disease, and chronic heart failure exacerbation. Hospitalization was defined as an overnight stay in the hospital and started with the admission date. Records that had the same admission and discharge date were excluded. Readmissions within 30 days were also excluded. The outcome definitions are provided in the S1 Table.

As described in the generic protocol [8], we accounted for the circulation of selected pathogens (RSV and influenza) using hospital-based viral proxies for community-wide viral activity derived from MDV. The latter have been shown to better predict hospitalization than surveillance-based proxies [21]. As in other model-based studies [7,10], the viral proxy for RSV circulation was defined as the number of weekly RSV-related hospitalizations (corresponding to ICD-10 codes: B34.8, J12.1, J20.5, J21.0, J21.9) in children younger than 2 years, due to the higher frequency of testing and hospitalization and higher sensitivity of diagnostic tests in this age group compared to adults [22]. We included acute bronchiolitis, unspecified (J21.9), because RSV is responsible for the majority of bronchiolitis hospitalizations in this age group [23,24] and was found to be a good indicator for the start of the RSV season [23–27]. The influenza proxy was defined as the number of influenza-specific hospitalizations (corresponding to ICD-10 codes: J09, J10, J11) in people 60 years and older.

## Statistical analysis and incidence calculation

The statistical analysis was conducted as in our published studies from Germany [17] and Spain [16] and according to the generic protocol [8] (Fig 1). In summary, we used quasi-Poisson regression to model the weekly number of hospitalizations for each subgroup (i.e., outcome per age group) as a function of periodic time trends, aperiodic time trends, and viral activity (RSV and influenza) while allowing for potential overdispersion. The model is given by:

$$Nr\_events_t \sim Poisson\left(\lambda_t, \theta\right)$$

with

$$\lambda_t = \left(\beta_0 + \sum_{k=1}^{4} \beta_k . t^k + \beta_5 . \sin\left(\frac{2t\pi}{52.143}\right) + \beta_6 . \cos\left(\frac{2t\pi}{52.143}\right) + \beta_7 . \sin\left(\frac{4t\pi}{52.143}\right) + \beta_8 . \cos\left(\frac{4t\pi}{52.143}\right)$$
$$+ \sum_{L=1}^{L} \beta_{(8+L)} . SVP_{l(t-m_l)}\right) * AllHosp_t$$

where $\lambda_t$ is the expected number of events in week t, with t representing the running index of the ISO weeks in the study period, and $\theta$ is the overdispersion parameter. Parameter $\beta_0$ is the coefficient associated with the baseline number of events, $\beta_1$ to $\beta_4$ are coefficients associated with the aperiodic time trend, $\beta_5$ to $\beta_8$ are coefficients associated with the periodic time trend and $\beta_9$ to $\beta_{(8+L)}$ are coefficients associated with the appropriately lagged scaled viral activity of the pathogen $SVP_l$, with l=1,2 reflecting the pathogen (RSV and influenza), and $m_l$=0,1,…,4 reflecting the pathogen-specific time lag. To account for the increase in the MDV database size, because the number of DPC hospitals providing data to the MDV database has increased over the years since its inception (from 16 hospitals in 2009–433 hospitals in 2019) [28], the total number of hospitalizations in week t (AllHospt) was added as an offset. We constructed the general models through a two-step model-building process and used the final subgroup-specific model to obtain the annual number of RSV-attributable hospitalizations for each subgroup. To account for the representativeness of the database, we projected the estimated number of RSV-attributable hospitalizations to all DPC hospitals in Japan using a year-specific scaling factor (i.e., number of all DPC hospitals in Japan/number of DPC hospitals in the MDV database) (derived from Japanese government statistics data [29]).

For each outcome and age group, annual IRs of RSV-attributable hospitalizations were obtained by dividing the projected yearly model-based age-specific number of RSV-attributable hospitalizations by the age-specific population at risk, obtained from annual national census data [30]. IRs are expressed as the number of hospitalizations per 100,000 person-years (PY). Confidence intervals for IRs were obtained using residual bootstrapping with 1,000 resampled datasets generated through sampling with replacement [8]. The IRs of RSV-attributable hospitalizations for individuals aged 60 years and older were obtained by summing the projected yearly model-based numbers of RSV-attributable hospitalizations for 60–79 years, and 80 years and older.

In Japan, the DPC hospitals account for approximately 54% of the total number of beds nationwide since 2015. To project the IRs of RSV-attributable DPC hospitals to all hospitals, while accounting for the uncertainty linked to structural differences between DPC and non-DPC hospitals, we carried out two different projections by multiplying the estimated IRs of RSV-attributable DPC hospitals by different sets of year-specific scaling factors. The first scaling factor is defined as the total number of beds in all hospitals divided by the total number of beds in the DPC hospitals [29]. The second scaling factor was calculated by comparing the outcome-specific hospitalizations obtained from the MDV database to those from the JMDC Inc. payer-based database, which covers all hospital types in Japan and is representative of working adults younger than 65 years [31]. For the purpose of deriving our scaling factor, we selected the JMDC age group 50–59 years, as it provided the most robust and representative data while closely aligning with the age distribution of our study population. These projections are based on the following assumptions: 1. For the first scaling factor: RSV burden is similar in adults aged 60 years and over admitted to the DPC and non-DPC hospitals; 2. For the second scaling factor: the ratio is comparable between adults aged 50–59 years, and 60 years and older.

In addition, to compare the difference between the observed (reported in the database regarding standard-of-care testing) versus the attributable (model-based) IR of RSV events, we also calculated the annual IR of RSV-specific hospitalizations (any of the following ICD-10 codes: J12.1, J20.5, J21.0) per age group.

We modeled the hospitalization data until 30th June 2019 to make use of the most recent available data (prior to the COVID-19 pandemic). However, we did not report the results for the year 2019 since the data are incomplete and the RSV peak occurred after 30th June that year (the RSV epidemic seasons 2017–2019 occurred between July and October in Japan); consequently, our model could not capture the vast majority of the RSV-attributable hospitalizations for the year 2019.

All statistical analyses were conducted in R version 4.2.3.

## Ethical considerations

The study followed generally accepted research practices described in the Good Epidemiological Practice guidelines issued by the International Epidemiological Association. According to the joint guidelines (latest revision 23 March 2021) of

MEXT (Ministry of Education, Culture, Sports, Science and Technology), MHLW (Ministry of Health, Labour and Welfare) and METI (Ministry of Economy, Trade and Industry), the current study does not require any ethics committee approval because the data derived from the MDV database used in the study only include anonymized de-identified/de-linked information.

## Results

### Observed hospitalizations

During the period from 1 January 2015 to 30 June 2019, a total of 1,116,457 cardiorespiratory DPC hospitalizations in adults 60 years and older were recorded in the MDV database, with yearly numbers ranging between 181,959 and 292,486 hospitalizations. Annual numbers of respiratory hospitalizations ranged between 97,487 and 151,326. The detailed number of hospitalizations is provided in S2 Table.

As described in Table 1, the annual IR of reported RSV-specific DPC hospitalizations based on RSV-specific ICD coding alone ranged from 0.03 to 0.1 cases per 100,000 PY in adults 60 years and older.

Data on RSV and influenza proxy are displayed in S3 Table and S1 Fig. Acute bronchiolitis, unspecified (J21.9), represented 5% of the RSV proxy counts.

### Estimated RSV-attributable hospitalizations

A seasonal pattern was observed for the two main outcomes. The model fitted well with the observed data in all covered age groups, as shown for the cardiorespiratory disease and the respiratory disease outcomes in Fig 2.

The estimated annual RSV-attributable DPC hospitalization IRs between 2015 and 2018 are reported in Table 2. Among individuals 60 years and older, the annual IRs of RSV-attributable cardiorespiratory hospitalizations at DPC hospitals ranged between 100–124 per 100,000 PY. The annual IRs of RSV-attributable respiratory hospitalizations at DPC hospitals in this age group were between 69 and 85 per 100,000 PY. The IRs remained stable across the study years, as evidenced by overlapping confidence intervals.

The highest RSV-attributable IRs for a secondary outcome were observed for influenza or pneumonia (37–46 DPC hospitalizations/100,000 PY in persons 60 years and older) followed by chronic heart failure exacerbation (11–14 DPC hospitalizations/100,000 PY in persons 60 years and older).

Regardless of the outcome of interest, the IRs were higher among adults 80 years and older (140–170 RSV-attributable cardiorespiratory DPC hospitalizations/100,000 PY) compared to adults aged 60–79 years (87–110 RSV-attributable cardiorespiratory DPC hospitalizations/100,000 PY).

As presented in Table 2, hospitalizations attributable to RSV represented up to 8% of the cardiorespiratory DPC hospitalizations and varied depending on the outcome and the age group between 1% to 12%. The proportion was higher among adults aged 60–79 years than among those 80 years and older for all outcomes.

**Table 1. Observed annual RSV-specific hospitalizations based on standard-of-care testing (Diagnoses recorded with RSV-specific ICD codes) incidence rates per 100,000 population by age group, 2015–2018, Japan.**

| Year | 60–79 years | ≥80 years | ≥60 years |
|------|-------------|-----------|-----------|
| 2015 | 0 | 0.1 | <0.1 |
| 2016 | 0 | 0.2 | 0.1 |
| 2017 | 0.1 | 0.2 | 0.1 |
| 2018 | 0.1 | 0.2 | 0.1 |

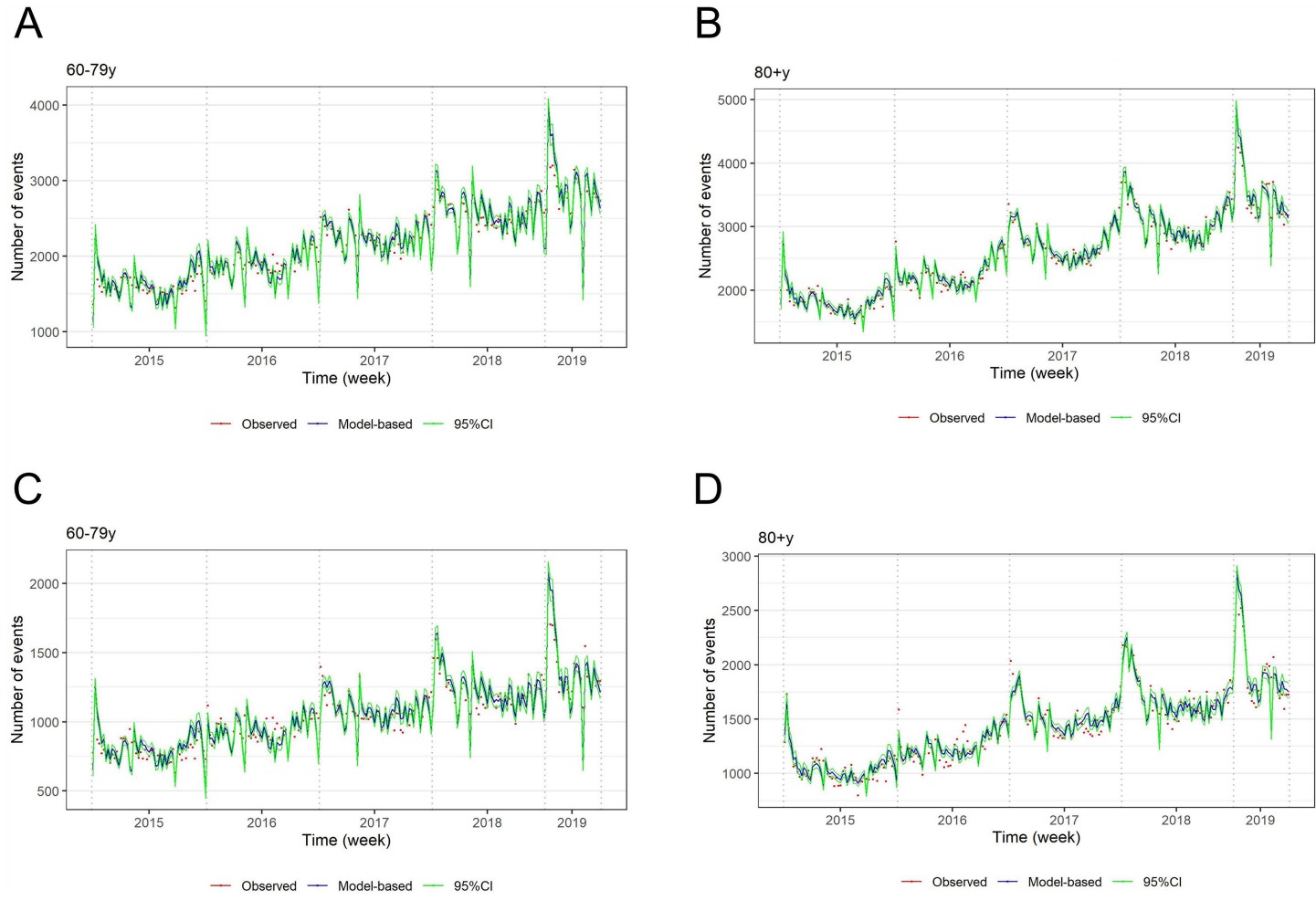

**Fig 2. Estimated vs observed cardiorespiratory (A and B) and respiratory (C and D) DPC hospitalizations in adults aged 60–79 and ≥80 years; Japan, 2015–2018.**

The projected annual IRs of RSV-attributable cardiorespiratory hospitalizations at all hospitals between 2015 and 2018 are reported in Table 3. Depending on the projection method, the annual IRs of RSV-attributable cardiorespiratory hospitalizations at all hospitals ranged between 134–229 cases per 100,000 PY and 96–157 RSV-attributable respiratory hospitalizations per 100,000 PY at all hospitals in individuals 60 years and older. The annual IRs were, on average, 1.8 and 1.4 times higher than the non-projected IRs, with the first and second scaling factors, respectively.

## Discussion

Our study establishes for the first time the RSV disease burden among older adults in Japan using a time-series model-based approach that accounts for undiagnosed RSV disease events. Our findings reveal a substantial hospitalization burden possibly attributed to RSV, with annual IRs ranging from 100–124 (DPC hospitals) to 134–229 cardiorespiratory hospitalizations per 100,000 PY when projected to all hospitals. Similarly, we estimated 69–85 (DPC hospitals) and 96–157 respiratory hospitalizations (all hospitals) per 100,000 PY in this age group. These results were substantially higher than the reported RSV burden among adults, indicating that our research has reaffirmed that the burden of RSV in

**Table 2. Estimated annual proportions (%) and incidence rates (IR) per 100,000 population of RSV-attributable cardiorespiratory DPC hospitalizations among older adults, projected to all DPC hospitals in Japan, 2015–2018.**

| Age groups | 60–79 years | | ≥80 years | | ≥60 years[†] | |
|---|---|---|---|---|---|---|
| Year | % | IR (95% CI) | % | IR (95% CI) | % | IR |
| **Cardiorespiratory diseases** | | | | | | |
| 2015 | 7.1 | 104.2 (68.0–141.0) | 3.1 | 169.5 (76.9–270.6) | 5.0 | 119.6 |
| 2016 | 5.6 | 86.8 (56.7–117.5) | 2.5 | 140.4 (63.7–224.0) | 3.9 | 99.8 |
| 2017 | 6.7 | 110.3 (72.1–149.4) | 2.8 | 165.7 (75.2–264.5) | 4.6 | 124.2 |
| 2018 | 5.9 | 103.1 (67.3–139.6) | 2.4 | 150.3 (68.2–239.9) | 4.0 | 115.2 |
| **Respiratory diseases** | | | | | | |
| 2015 | 8.2 | 59.8 (40.0–81.8) | 4.9 | 154.7 (83.2–234.3) | 6.3 | 82.1 |
| 2016 | 6.5 | 49.8 (33.3–68.1) | 4.1 | 128.1 (68.9–194.0) | 5.1 | 68.8 |
| 2017 | 8.0 | 63.3 (42.4–86.6) | 4.6 | 151.3 (81.3–229.1) | 6.0 | 85.3 |
| 2018 | 7.1 | 59.1 (39.6–80.9) | 4.0 | 137.1 (73.7–207.7) | 5.3 | 79.1 |
| **Influenza or pneumonia** | | | | | | |
| 2015 | 11.6 | 34.4 (21.6–48.4) | 5.7 | 77.1 (35.1–124.0) | 8.2 | 44.5 |
| 2016 | 8.9 | 28.7 (18.0–40.3) | 4.6 | 63.8 (29.0–102.6) | 6.4 | 37.2 |
| 2017 | 11.1 | 36.5 (22.8–51.2) | 5.1 | 75.4 (34.3–121.2) | 7.5 | 46.2 |
| 2018 | 9.5 | 34.1 (21.3–47.9) | 4.3 | 68.3 (31.1–109.9) | 6.4 | 42.9 |
| **Chronic lower respiratory diseases** | | | | | | |
| 2015 | 7.2 | 5.3 (2.1–8.9) | 4.7 | 9.4 (0.3–17.7) | 6.0 | 6.2 |
| 2016 | 5.7 | 4.4 (1.7–7.4) | 3.8 | 7.8 (0.2–14.7) | 4.8 | 5.2 |
| 2017 | 6.6 | 5.6 (2.2–9.4) | 4.2 | 9.2 (0.3–17.4) | 5.4 | 6.5 |
| 2018 | 6.0 | 5.2 (2.0–8.8) | 3.7 | 8.3 (0.3–15.7) | 4.9 | 6.0 |
| **Chronic heart failure exacerbation** | | | | | | |
| 2015 | 5.0 | 12.2 (4.3–20.1) | 1.2 | 15.9 (0–44.2) | 2.7 | 13.1 |
| 2016 | 4.2 | 10.2 (3.6–16.7) | 1.0 | 13.2 (0–36.6) | 2.1 | 10.9 |
| 2017 | 4.7 | 13.0 (4.6–21.3) | 1.0 | 15.5 (0–43.2) | 2.3 | 13.6 |
| 2018 | 4.1 | 12.1 (4.3–19.9) | 0.9 | 14.1 (0–39.2) | 2.0 | 12.6 |

[†]based on the pooling of results for 60–79 and ≥80 years.

Note: we modeled data from 1 January 2015 to 30 June 2019 but report only the results for complete years (2015–2018).

Japan has been underestimated. While the persons 60–79 years exhibited a higher proportion of RSV-attributable hospitalizations, their IRs were lower compared to those 80 years and older. This likely reflects differences in population size and competing risks. Individuals 80 years and older are more likely to be hospitalized for non-RSV reasons, which dilutes the proportion of hospitalizations specifically attributed to RSV, even if their risk of infection relative to the population size (IR) is higher. The increased IR trend with age was expected and consistent with results obtained in other countries [16,17]; this is likely due to several factors, including the natural decline of the immune response (immunosenescence), the chronic increase in basal systemic inflammation (inflammaging), and the increased prevalence of comorbidities with age [32].

In addition to the typical respiratory conditions associated with RSV, we found that RSV contributes also to cardiovascular conditions, as hospitalization rates are 1.5 times higher for cardiorespiratory disease than for respiratory disease alone. It is well known that RSV can exacerbate pre-existing cardiovascular diseases such as chronic heart failure via either direct effect on the myocardium or indirect effects originating from the infected respiratory tract (e.g., due to pulmonary hypertension secondary to severe RSV bronchitis), acute hypoxia, or inflammation [33]. Past research also

**Table 3. Estimated incidence rates (IRs) of RSV-attributable cardiorespiratory and respiratory hospitalizations in adults aged ≥60 years, projected to all hospitals in Japan, 2015–2018.**

| Age groups | | 60–79 years | ≥80 years | ≥60 years† |
|---|---|---|---|---|
| Year | Scaling factor | IR (95%CI) | IR (95%CI) | IR |
| **Cardiorespiratory disease** | | | | |
| **Method 1** | | | | |
| 2015 | 1.9 | 194.8 (127.2–263.7) | 317.0 (143.9–506.0) | 223.6 |
| 2016 | 1.8 | 156.2 (102.0–211.5) | 252.7 (114.7–403.3) | 179.7 |
| 2017 | 1.8 | 203.0 (132.6–274.9) | 305.0 (138.4–486.8) | 228.6 |
| 2018 | 1.8 | 187.6 (122.5–254.0) | 273.5 (124.2–436.6) | 209.6 |
| **Method 2** | | | | |
| 2015 | 1.3 | 137.5 (89.8–186.2) | 223.8 (101.6–357.2) | 157.8 |
| 2016 | 1.3 | 116.3 (76.0–157.0) | 188.1 (85.4–300.2) | 133.8 |
| 2017 | 1.3 | 144.5 (94.4–195.7) | 217.1 (98.6–346.6) | 162.7 |
| 2018 | 1.3 | 132.0 (86.2–178.7) | 192.4 (87.3–307.0) | 147.4 |
| **Respiratory disease** | | | | |
| **Method 1** | | | | |
| 2015 | 1.9 | 111.7 (74.9–152.9) | 289.3 (155.6–438.3) | 153.5 |
| 2016 | 1.8 | 89.6 (60.0–122.6) | 230.6 (124.0–349.3) | 123.9 |
| 2017 | 1.8 | 116.5 (78.0–159.4) | 278.3 (149.7–421.6) | 157.0 |
| 2018 | 1.8 | 107.6 (72.1–147.3) | 249.6 (134.2–378.1) | 144.0 |
| **Method 2** | | | | |
| 2015 | 1.4 | 80.7 (54.0–110.4) | 208.9 (112.3–316.4) | 110.9 |
| 2016 | 1.4 | 69.7 (46.7–95.4) | 179.3 (96.4–271.7) | 96.4 |
| 2017 | 1.4 | 88.0 (58.9–120.4) | 210.3 (113.1–318.5) | 118.6 |
| 2018 | 1.4 | 82.2 (55.1–112.5) | 190.6 (102.5–288.8) | 110.0 |

†based on the pooling of results for 60–79 and ≥80 years.

Method 1: number of beds in DPC-eligible hospitals vs. number of beds in all hospitals in Japan; Method 2: number of outcome-specific hospitalizations in 50–59y age group in JMDC/ number of outcome-specific hospitalizations in 50–59y age group in MDV.

Note: we modeled data from 1 January 2015 to 30 June 2019 but report only the results for complete years (2015–2018).

demonstrated that RSV can trigger new cardiovascular events [20,21]. Unfortunately, we were not able to model all cardiovascular diseases because this outcome did not present clear seasonal patterns; however, we estimated that 1–5% of chronic heart failure exacerbation hospitalizations in adults aged 60–79 years could be attributed to RSV.

Of the three outcome subgroups modeled, the highest IRs among older adults were found for the influenza or pneumonia diagnosis grouping (37–46 DPC hospitalizations per 100,000 PY in adults 60 years and older). RSV is now being recognized as one of the leading causes of acute respiratory infections in older adults, including among patients with pneumonia [12]. RSV pneumonia can present with imaging features that are similar to other infectious pneumonias, making it difficult to differentiate from other causes of infectious pneumonia based on radiographic imaging [34–39].

In general, the hospitalization burden can vary across countries because of several factors, such as access to care, hospitalization practices, and geographical variety. The rates of RSV-attributable respiratory hospitalizations (69–85 DPC hospitalizations per 100,000 PY) in Japan are consistently lower than those from studies conducted in parallel for other countries using the same model, such as Spain (257–283 hospitalizations per 100,000 PY in 60 years and older) [16]. These differences are observed across all outcomes and age groups. This could have occurred for several reasons. For example, although our source database is representative of DPC hospitals in Japan and most likely covers the majority of the RSV-attributable hospitalizations, it might not be fully representative of less acute cases since DPC hospitals focus on

advanced treatment. Therefore, we projected IRs to all Japanese hospitals using two different methods. The range of the projected IRs (96–157 respiratory hospitalizations per 100,000 PY in 60 years and older) is closer to the IRs obtained in the other countries. They are also in line with the results from a similar modeling study conducted in the UK for the annual rates of RSV-attributable respiratory (62–101 and 180–291 hospitalizations per 100,000 PY in persons aged 65–74 years and in 75 years and older, respectively) and pneumonia/influenza hospitalizations (15–26 and 51–109 hospitalizations per 100,000 PY in persons age 65–74 years and in 75 years and older, respectively) [7].

Our study had several limitations. First, all Japanese regions were pooled into one dataset for modeling, while different regions are likely to display different RSV seasonality patterns (e.g., Okinawa region has a subtropical climate [40]). Unfortunately, we were not able to account for this by stratifying by region, as no appropriate geographical indicator was available in the source data. For the same reason, we also did not stratify the burden of RSV by comorbidities, which are known to significantly influence disease severity and outcomes in older adults. We also did not include proxies for the activity of other respiratory pathogens in our model, which might have resulted in an overestimation of RSV burden. However, this approach has been successfully used in studies conducted in other countries [16,17,41,42], and even without explicitly modeling other potentially relevant pathogens, they are largely accounted for in the model indirectly through the periodic component and the overdispersion parameter. Our projections relied heavily on several assumptions that could not be verified concerning the consistency of the distribution of RSV burden across the different hospital types (MDV/non-MDV DPC hospitals, DPC/non-DPC hospitals) and age groups (50–59 years/60 years and older). Given the inability to validate these assumptions with external data, we chose to present estimate ranges to reflect the associated uncertainty. We did not model the data for 18–60 years due to the low case counts or to the absence of clearly distinguishable seasonality. Our study focused on individuals aged 60 years and older who are at the highest risk of severe disease, making it critical to quantify the disease burden for them to guide public health strategies. Lastly, the year 2019 was excluded from the final estimates due to incomplete data for that season, however the partial data available for 2019 did not show any deviation from the model's predictions. Specifically, IRs for 2019 were consistent with the trends observed in the other years, as indicated by overlapping confidence intervals (data not shown).

Our study also had several strengths. It is the first model-based study that estimated RSV-associated cardiorespiratory hospitalization burden among older adults in Japan. We used a database that includes DPC hospitalizations across different regions throughout Japan, which reduced the possibility of sampling bias. We also used an extensive list of ICD-10 codes, corresponding to broad definitions for respiratory outcomes and more specific for cardiovascular diseases, thereby improving the sensitivity of detecting RSV-related outcomes [16,43–45].

Our study provides unique and useful data to better assess the RSV disease burden among the growing population of older adults in Japan and to inform decisions concerning RSV prevention, particularly the use of newly developed RSV vaccines. Our data indicate that future studies of RSV burden and preventable burden need to consider both respiratory and cardiovascular outcomes. Future studies should include assessing vaccine-preventable burden among adults with comorbidities and against a broad group of outcomes. They should also assess non-hospitalization burden, mortality, and longer-term consequences of more severe RSV infection, such as impact on quality of life, requirements for assisted living facilities, and functioning of extended family members serving as caretakers.

## Supporting information

**S1 Table. Definition of hospitalization main and secondary outcomes.**
(DOCX)

**S2 Table. Annual number of hospitalizations due to cardiorespiratory diseases (based on the source data [non-extrapolated]) by age group, January 2015−June 2019, Japan.**
(DOCX)

**S3 Table. Annual number of RSV and influenza proxy DPC (Diagnosis Procedure Combination) hospitalizations (based on the source data [non-projected]), January 2015−June 2019, Japan.**
(DOCX)

**S1 Fig. Weekly number of RSV and influenza proxy DPC hospitalizations (based on the source data [non-projected]), January 2015−June 2019, Japan.**
(DOCX)

## Acknowledgments

We would like to acknowledge Worku Ewnetu, a statistician at P95, for his support during the statistical analysis. Editorial support was provided by Somsuvro Basu of P95.

## Author contributions

**Conceptualization:** Masafumi Seki, Aleksandra Polkowska-Kramek, Caihua Liang, Robin Bruyndonckx, Bradford D. Gessner, Elizabeth Begier.

**Data curation:** Yasuhiro Kobayashi, Estelle Méroc, Takahiro Kitano, Aleksandra Polkowska-Kramek, Robin Bruyndonckx, Solomon Molalign Moges, Eduardo Conde-Sousa.

**Formal analysis:** Estelle Méroc, Robin Bruyndonckx, Solomon Molalign Moges, Eduardo Conde-Sousa.

**Methodology:** Masafumi Seki, Yasuhiro Kobayashi, Estelle Méroc, Takahiro Kitano, Aleksandra Polkowska-Kramek, Asuka Yoshida, Caihua Liang, Robin Bruyndonckx, Charles Nuttens, Bradford D. Gessner, Elizabeth Begier.

**Project administration:** Estelle Méroc, Aleksandra Polkowska-Kramek, Caihua Liang, Elizabeth Begier.

**Software:** Estelle Méroc, Robin Bruyndonckx, Solomon Molalign Moges, Eduardo Conde-Sousa.

**Supervision:** Estelle Méroc, Aleksandra Polkowska-Kramek, Caihua Liang, Robin Bruyndonckx, Elizabeth Begier.

**Validation:** Masafumi Seki, Yasuhiro Kobayashi, Estelle Méroc, Takahiro Kitano, Aleksandra Polkowska-Kramek, Robin Bruyndonckx, Solomon Molalign Moges, Eduardo Conde-Sousa.

**Visualization:** Estelle Méroc, Solomon Molalign Moges.

**Writing – original draft:** Estelle Méroc, Aleksandra Polkowska-Kramek, Robin Bruyndonckx.

**Writing – review & editing:** Masafumi Seki, Yasuhiro Kobayashi, Estelle Méroc, Takahiro Kitano, Aleksandra Polkowska-Kramek, Asuka Yoshida, Shuhei Ito, Caihua Liang, Robin Bruyndonckx, Solomon Molalign Moges, Eduardo Conde-Sousa, Charles Nuttens, Bradford D. Gessner, Elizabeth Begier.

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
