## [Decision Letter · Decision Letter 0]

25 Jul 2025

Dear Dr. Liang,

Thank you for submitting your manuscript to PLOS ONE. After careful consideration, we feel that it has merit but does not fully meet PLOS ONE’s publication criteria as it currently stands. Therefore, we invite you to submit a revised version of the manuscript that addresses the points raised during the review process.

We look forward to receiving your revised manuscript.

Kind regards,

Liling Chaw

Academic Editor

PLOS ONE

Journal Requirements:

This study was sponsored by Pfizer Inc.

We would like to acknowledge Worku Ewnetu, a statistician at P95, for his support during the statistical analysis. Editorial support was provided by Somsuvro Basu of P95 and was funded by Pfizer.

This study was sponsored by Pfizer Inc.

4. Thank you for stating the following in the Competing Interests/Financial Disclosure section: I have read the journal's policy and the authors of this manuscript have the following competing interests:

Caihua Liang, Yasuhiro Kobayashi, Takahiro Kitano, Asuka Yoshida, Shuhei Ito, Bradford D. Gessner, Charles Nuttens, and Elizabeth Begier are employees of Pfizer and may own Pfizer stock. Estelle Méroc, Solomon Molalign, Eduardo Conde-Sousa, Robin Bruyndonckx, and Aleksandra Polkowska-Kramek are employees of P95, which received funding from Pfizer in connection with the development of this manuscript and conducting this study.

We note that one or more of the authors are employed by commercial companies: Pfizer and P95

5. We note that you have indicated that there are restrictions to data sharing for this study. For studies involving human research participant data or other sensitive data, we encourage authors to share de-identified or anonymized data. However, when data cannot be publicly shared for ethical reasons, we allow authors to make their data sets available upon request. For information on unacceptable data access restrictions, please see http://journals.plos.org/plosone/s/data-availability#loc-unacceptable-data-access-restrictions.

Reviewers' comments:

Reviewer's Responses to Questions

**Comments to the Author**

1. Is the manuscript technically sound, and do the data support the conclusions?

Reviewer #1: Yes

Reviewer #2: Yes

2. Has the statistical analysis been performed appropriately and rigorously?

Reviewer #1: Yes

Reviewer #2: I Don't Know

3. Have the authors made all data underlying the findings in their manuscript fully available?

Reviewer #1: Yes

Reviewer #2: No

4. Is the manuscript presented in an intelligible fashion and written in standard English?

Reviewer #1: Yes

Reviewer #2: Yes

Reviewer #1: This is an automated report for PONE-D-25-13816. This report was solicited by the PLOS One editorial team and provided by ScreenIT.

ScreenIT is an independent group of scientists developing automated tools that analyze academic papers. A set of automated tools screened your submitted manuscript and provided the report below. Each tool was created by your academic colleagues with the goal of helping authors. The tools look for factors that are important for transparency, rigor and reproducibility, and we hope that the report might help you to improve reporting in your manuscript. Within the report you will find links to more information about the items that the tools check. These links include helpful papers, websites, or videos that explain why the item is important. While our screening tools aim to improve and maintain quality standards they may, on occasion, miss nuances specific to your study type or flag something incorrectly. Each tool has limitations that are described on the ScreenIT website. The tools screen the main file for the paper; they are not able to screen supplements stored in separate files. Please note that the Academic Editor had access to these comments while making a decision on your manuscript. The Academic Editor may ask that issues flagged in this report be addressed. If you would like to provide feedback on the ScreenIT tool, please email the team at ScreenIt@bih-charite.de. If you have questions or concerns about the review process, please contact the PLOS One office at plosone@plos.org.

Reviewer #2: This manuscript addresses a significant research gap by utilizing a representative sample and providing valuable population statistics. However, the statistical methodology could be clarified, particularly regarding the specific time series model employed and its formulation. Additionally, the manuscript would benefit from descriptive analysis, including comparisons across socio-demographic variables like gender. Despite these limitations, the work is noteworthy and contributes meaningfully to the field.

**Do you want your identity to be public for this peer review?** For information about this choice, including consent withdrawal, please see our Privacy Policy

Reviewer #1: No

Reviewer #2: **Yes:** Oluchukwu Chiamaka Okorie

---

## [Author Response · Author response to Decision Letter 1]

13 Oct 2025

Dr. Liling Chaw

Academic Editor

PLOS ONE

Dear Dr. Chaw,

We are grateful for the opportunity to revise the manuscript in light of the reviewers’ comments and journal requirements. We have responded to all comments as itemized in our point-by-point reply below in blue. All changes are saved in tracked changes in the revised version (file name with ending: Manuscript_with_Track_Changes) of our manuscript. An unmarked version of your revised paper without tracked changes (file name with ending: Manuscript).

Sincerely yours,

Caihua Liang

on behalf of the authors

-------------

Journal Requirements:

and

Response: We have formatted the revised manuscript following PLOS ONE's style requirements.

This study was sponsored by Pfizer Inc.

Response: We have amended the Funding Statement, stated below.

“This study was sponsored by Pfizer Inc. Several authors [EM, SM, ECS, RB, and APK] are employees of P95, which received funding from Pfizer Inc. for the conduct of this study, and for the development and editorial support of this manuscript.

The funder provided support in the form of salaries for some authors [YK, TK, AY, SI, CL, CN, BDG, and EB], but non-authors from Pfizer did not have any additional role in the study design, data collection and analysis, decision to publish, or preparation of the manuscript. The specific roles of these authors are articulated in the ‘author contributions’ section.”

We would like to acknowledge Worku Ewnetu, a statistician at P95, for his support during the statistical analysis. Editorial support was provided by Somsuvro Basu of P95 and was funded by Pfizer.

This study was sponsored by Pfizer Inc.

Response: We have amended the Acknowledgments Section (information on funding for editorial support has been moved to the funding statement) and Funding Statement accordingly.

Acknowledgments Section:

“We would like to acknowledge Worku Ewnetu, a statistician at P95, for his support during the statistical analysis. Editorial support was provided by Somsuvro Basu of P95.”

Funding statement:

“This study was sponsored by Pfizer Inc. Several authors [EM, SM, ECS, RB, and APK] are employees of P95, which received funding from Pfizer Inc. for the conduct of this study, and for the development and editorial support of this manuscript.

The funder provided support in the form of salaries for some authors [YK, TK, AY, SI, CL, CN, BDG, and EB], but non-authors from Pfizer did not have any additional role in the study design, data collection and analysis, decision to publish, or preparation of the manuscript. The specific roles of these authors are articulated in the ‘author contributions’ section.”

4. Thank you for stating the following in the Competing Interests/Financial Disclosure section: I have read the journal's policy and the authors of this manuscript have the following competing interests:

Caihua Liang, Yasuhiro Kobayashi, Takahiro Kitano, Asuka Yoshida, Shuhei Ito, Bradford D. Gessner, Charles Nuttens, and Elizabeth Begier are employees of Pfizer and may own Pfizer stock. Estelle Méroc, Solomon Molalign, Eduardo Conde-Sousa, Robin Bruyndonckx, and Aleksandra Polkowska-Kramek are employees of P95, which received funding from Pfizer in connection with the development of this manuscript and conducting this study.

We note that one or more of the authors are employed by commercial companies: Pfizer and P95

Response: We have amended the Funding Statement accordingly, declaring the commercial affiliation of the authors, as well as a statement regarding the Role of the Funder in the study.

Amended Funding Statement:

“This study was sponsored by Pfizer Inc. Several authors [EM, SM, ECS, RB, and APK] are employees of P95, which received funding from Pfizer Inc. for the conduct of this study, and for the development and editorial support of this manuscript.

The funder provided support in the form of salaries for some authors [YK, TK, AY, SI, CL, CN, BDG, and EB], but non-authors from Pfizer did not have any additional role in the study design, data collection and analysis, decision to publish, or preparation of the manuscript. The specific roles of these authors are articulated in the ‘author contributions’ section.”

The Author Contributions section remains unchanged, but the authors' names are formatted to initials, as stated below.

“BDG and EB conceived the study. MS, LC, RB, APK, and EB designed the study and its statistical analysis. EM, SM, and ECS conducted the statistical analysis. MS, EM, RB, and APK wrote the manuscript, with CL, YK, TK, AY, SI, BDG, CN, and EB providing critical feedback and revisions. All authors reviewed and approved the final version of the manuscript.”

b. Please also provide an updated Competing Interests Statement declaring this commercial affiliation along with any other relevant declarations relating to employment, consultancy, patents, products in development, or marketed products, etc .

Response: We have updated the Competing Interests Statement, as stated below:

“P95 received funding from Pfizer Inc. for the conduct of this study, and for the development and editorial support of this manuscript. Caihua Liang, Yasuhiro Kobayashi, Takahiro Kitano, Asuka Yoshida, Shuhei Ito, Bradford D. Gessner, Charles Nuttens, and Elizabeth Begier are employees of Pfizer and may own Pfizer stock. Estelle Méroc, Solomon Molalign, Eduardo Conde-Sousa, Robin Bruyndonckx, and Aleksandra Polkowska-Kramek are employees of P95. This does not alter the authors’ adherence to PLOS ONE policies on sharing data and materials.”

5. We note that you have indicated that there are restrictions to data sharing for this study. For studies involving human research participant data or other sensitive data, we encourage authors to share de-identified or anonymized data. However, when data cannot be publicly shared for ethical reasons, we allow authors to make their data sets available upon request. For information on unacceptable data access restrictions, please see http://journals.plos.org/plosone/s/data-availability#loc-unacceptable-data-access-restrictions.

Response: The Data Availability Statement has been revised to include details regarding data access restrictions and information about the data owner.

Updated Data Availability Statement:

“The datasets generated during and/or analyzed during the current study are not publicly available. The data are owned by Medical Data Vision (MDV) Co., Ltd, and the authors do not have permission to share the data. The data are available upon request to MDV (https://en.mdv.co.jp). The statistical codes are available upon request to the corresponding author.”

Response: Not applicable

Response: The references are checked and formatted following the PLOS ONE’s style requirements.

Comments to the Author

1. Is the manuscript technically sound, and do the data support the conclusions?

Reviewer #1: Yes

Reviewer #2: Yes

Response: We thank the reviewers for their confirmation.

2. Has the statistical analysis been performed appropriately and rigorously?

Reviewer #1: Yes

Reviewer #2: I Don't Know

Response: We thank the reviewers for their confirmation. We performed a quality check that involved double programming of the statistical analyses to ensure accuracy and reliability in our results. Our code could also be made available upon request. ________________________________________

3. Have the authors made all data underlying the findings in their manuscript fully available?

Reviewer #1: Yes

Reviewer #2: No

Response: We thank the reviewers.

The datasets generated and/or analyzed during the current study are not publicly available. The data are owned by Medical Data Vision (MDV) Co., Ltd, and the authors do not have permission to share the data. The data are available upon request to MDV (https://en.mdv.co.jp) as clarified in the data availability statement.

4. Is the manuscript presented in an intelligible fashion and written in standard English?

Reviewer #1: Yes

Reviewer #2: Yes

Response: We thank the reviewers for their confirmation.

5. Review Comments to the Author

Reviewer #1:

This is an automated report for PONE-D-25-13816. This report was solicited by the PLOS One editorial team and provided by ScreenIT.

ScreenIT is an independent group of scientists developing automated tools that analyze academic papers. A set of automated tools screened your submitted manuscript and provided the report below. Each tool was created by your academic colleagues with the goal of helping authors. The tools look for factors that are important for transparency, rigor and reproducibility, and we hope that the report might help you to improve reporting in your manuscript. Within the report you will find links to more information about the items that the tools check. These links include helpful papers, web

---

## [Decision Letter · Decision Letter 1]

2 Dec 2025

Dear Dr. Liang,

Thank you for submitting your manuscript to PLOS ONE. After careful consideration, we feel that it has merit but does not fully meet PLOS ONE’s publication criteria as it currently stands. Therefore, we invite you to submit a revised version of the manuscript that addresses the points raised during the review process.

We look forward to receiving your revised manuscript.

Kind regards,

Liling Chaw

Academic Editor

PLOS ONE

Journal Requirements:

Reviewers' comments:

Reviewer's Responses to Questions

**Comments to the Author**

Reviewer #2: All comments have been addressed

Reviewer #3: (No Response)

2. Is the manuscript technically sound, and do the data support the conclusions?

Reviewer #2: Yes

Reviewer #3: Partly

3. Has the statistical analysis been performed appropriately and rigorously?

Reviewer #2: Yes

Reviewer #3: Yes

4. Have the authors made all data underlying the findings in their manuscript fully available?

Reviewer #2: No

Reviewer #3: No

5. Is the manuscript presented in an intelligible fashion and written in standard English?

Reviewer #2: Yes

Reviewer #3: Yes

Reviewer #2: The authors have addressed the comments raised in the methodology by naming the time series model that was used.They have also added a table of the descriptives too.

Reviewer #3: I read this article on an important and timely topic. The Results present valuable numeric estimates. Significant revisions (some of which have already been mentioned in the discussion) are required to clarify the methodology, justify key assumptions, and validate data sources. Contextualization relative to national level is insufficient. I recommend major revision.

Introduction

L74: If available, report the actual proportion of adults aged 60+ years in Japan instead of stating “a large proportion.”

L75: specify which new vaccines are available for adults aged +60 years.

Methods

L114-117: Some important cardiovascular categories are not included.

L124-128: The choice of J21.9 as part of the RSV proxy for children <2 is not supported by Japanese validation data. The RSV proportion for this population category has been shown to be around 60% and is highly variable (doi: 10.1371/journal.pone.0242302).

L128-129: The influenza proxy is based on hospitalizations in adults +65 years, although the study population is +60 years.

The manuscript does not describe the source, reliability, or testing practices underlying RSV and Influenza proxies.

L179-184: Excluding an entire year because data are incomplete could be suboptimal. A sensitivity analysis including partial 2019 data could strengthen the robustness of the results.

The 2015-2018 window (4 years) is relatively short.

L153-155: The scaling factor assumes DPC and non-DPC hospitals are comparable, which is unlikely (differences in patient age, specialization…).

L.171-174: The manuscript does not discuss known structural differences between DPC and non-DPC hospitals (e.g., geriatric, rehabilitation facilities), which could significantly affect RSV burden estimates.

L168-170: The second scaling method, using adults aged 50-59 years, relies on an unproven assumption that patterns in 50-59 years reflect those +60. This needs justification.

L159-160: The residual bootstrap procedure lacks details (number of replicates, resampling method…).

No stratification by comorbidities.

No stratification by Japanese regions.

No analysis of mortality.

No data on 18-60 years.

No non-RSV/Influenza proxies.

Results

Lines 198-202: You report a total of 1.1 million cardiorespiratory hospitalizations in the MDV database, is it possible to contextualize these numbers in relation to national hospitalization statistics. Without such context, readers cannot evaluate whether the MDV case load is proportionally higher or lower than expected.

Although Table 2 displays year-specific incidence rates, the Results section does not discuss inter-annual fluctuations.

You exclude 2019 from final estimates, but the Results do not discuss whether the partial season shows deviations or supports the model’s predictive behaviour.

Discussion

Higher proportions in 60-79y and higher IRs in ≥80y could be further explained.

Lines 297-300: confidence intervals could be provided to strengthen the robustness of international comparisons.

Projections rely on unverifiable assumptions regarding hospital types and age groups, which may under- or overestimate burden. Validation with external datasets would have been valuable.

Future studies should consider 18+ years, mortality...

**Do you want your identity to be public for this peer review?** For information about this choice, including consent withdrawal, please see our Privacy Policy

Reviewer #2: **Yes:** Oluchukwu Okorie

Reviewer #3: No

---

## [Author Response · Author response to Decision Letter 2]

8 Jan 2026

Dr. Liling Chaw

Academic Editor

PLOS ONE

Date: 08 January 2026

Dear Dr. Chaw,

We are grateful for the opportunity to revise the manuscript in light of the reviewers’ comments and journal requirements. We have responded to all comments as itemized in our point-by-point reply below in blue. All changes are saved in tracked changes in the revised version (file name with: Revised_Manuscript_with_Track_Changes) of our manuscript. An unmarked version of our revised manuscript without tracked changes (file name with: Manuscript) is also shared.

Sincerely yours,

Caihua Liang

on behalf of the authors

Journal Requirements:

Response: Not applicable. The inclusion of the newly cited publications (Refs. 21, 24-27, 35-39, 41, 42) was necessary for the updated version of the manuscript.

Response: Not applicable. This has been reviewed and verified.

Reviewers' comments:

Reviewer's Responses to Questions

Comments to the Author

1. If the authors have adequately addressed your comments raised in a previous round of review and you feel that this manuscript is now acceptable for publication, you may indicate that here to bypass the “Comments to the Author” section, enter your conflict of interest statement in the “Confidential to Editor” section, and submit your "Accept" recommendation.

Reviewer #2: All comments have been addressed

Reviewer #3: (No Response)

Response: We thank reviewer #2 for the confirmation.

2. Is the manuscript technically sound, and do the data support the conclusions?

Reviewer #2: Yes

Reviewer #3: Partly

Response: We thank the reviewers for their (partial) confirmation. We have addressed the concerns raised by Reviewer #3; please find our detailed answers under section 6.

3. Has the statistical analysis been performed appropriately and rigorously?

Reviewer #2: Yes

Reviewer #3: Yes

Response: We thank the reviewers for their confirmation.

4. Have the authors made all data underlying the findings in their manuscript fully available?

Reviewer #2: No

Reviewer #3: No

Response: The Data Availability Statement was updated in the last revision round to explain why the datasets are not publicly available and how they can be requested. We share the Data Availability Statement, again below:

Data Availability Statement: “The datasets generated during and/or analyzed during the current study are not publicly available. The data are owned by a third-party, Medical Data Vision (MDV) Co., Ltd, and the authors do not have permission to share the data. The data are available upon request to MDV (https://en.mdv.co.jp; contacts: Hiroyuki Tada [email: tada@mdv.co.jp] and Yuki Santo [email: santo_yuki@mdv.co.jp]).

The statistical codes are available upon request to the following contacts: statistics department, P95 (email: stat@p-95.com), and Hannah Volkman (Pfizer Inc; email: Hannah.Volkman@pfizer.com).”

5. Is the manuscript presented in an intelligible fashion and written in standard English?

Reviewer #2: Yes

Reviewer #3: Yes

Response: We thank the reviewers for their confirmation.

6. Review Comments to the Author

Reviewer #2:

The authors have addressed the comments raised in the methodology by naming the time series model that was used. They have also added a table of the descriptives too.

Response: Thank you for taking the time to review our work. We truly appreciate your thoughtful feedback and support.

-------------

Reviewer #3:

I read this article on an important and timely topic. The Results present valuable numeric estimates. Significant revisions (some of which have already been mentioned in the discussion) are required to clarify the methodology, justify key assumptions, and validate data sources. Contextualization relative to national level is insufficient. I recommend major revision.

Introduction

• L74: If available, report the actual proportion of adults aged 60+ years in Japan instead of stating “a large proportion.”

Response: Thank you for your comment. We have added the following information: “…representing 41% in 2019 [14]” (line 75 in the clean revised manuscript).

• L75: specify which new vaccines are available for adults aged +60 years.

Response: We have added the following information: “..such as the protein subunit vaccines and mRNA vaccine,” (lines 76-77 in the clean revised manuscript).

Methods

• L114-117: Some important cardiovascular categories are not included.

Response: As described in our generic protocol (1), we have included the selected cardiovascular disease codes most likely to be associated with RSV, as previously reported in the literature (2-9) and used in other model-based studies.

1) Bruyndonckx R, Polkowska-Kramek A, Liang C, Nuttens C, Tran TMP, Gessner BD, et al. Estimation of symptomatic respiratory syncytial virus infection incidence in adults in multiple countries: a time-series model-based analysis protocol. Infectious Diseases and Therapy. 2024;13(4):953-63. doi: 10.1007/s40121-024-00948-9.

2) Kujawski SA, Whitaker M, Ritchey MD, Reingold AL, Chai SJ, Anderson EJ, et al. Rates of respiratory syncytial virus (RSV)-associated hospitalization among adults with congestive heart failure-United States, 2015–2017. PLoS ONE. 2022;17(3): e0264890.

3) Ivey KS, Edwards KM, Talbot HK. Respiratory syncytial virus and associations with cardio vascular disease in adults. J Am Coll Cardiol. 2018;71(14):1574–83.

4) Franczuk P, Tkaczyszyn M, Kulak M, Domenico E,Ponikowski P, Jankowska EA. Cardiovascular complications of viral respiratory infections and COVID-19. Biomedicines. 2022. https://doi.org/10.3390/biomedicines11010071.

5) Zhou H, Thompson WW, Viboud CG, Ringholz CM, Cheng PY, Steiner C, et al. Hospitalizations associated with influenza and respiratory syncytial virus in the United States, 1993–2008. Clin Infect Dis. 2012;54(10):1427–36

6) Matias G, Taylor RJ, Haguinet F, Schuck-Paim C, Lustig RL, Shinde V. Estimates of mortality attributable to influenza and RSV in the United States during 1997–2009 by influenza type or subtype, age, cause of death, and risk status. Influenza Other Respir Viruses. 2014;8(5):507–15

7) Mullooly J, Bridges C, Thompson W, Chen J, Weintraub E, Jackson L, et al. Influenza- and RSV-associated hospitalizations among adults. Vaccine.2007;25(5):846–55.

8) Kawashima H, Inagaki N, Nakayama T, Morichi S, Nishimata S, Yamanaka G, et al. Cardiac complications caused by respiratory syncytial virus infection: questionnaire survey and a literature review. Glob Pediatr Health. 2021. https://doi.org/10.1177/ 2333794X211044114

9) Corrales-Medina VF, Madjid M, Musher DM. Role of acute infection in triggering acute coronary syndromes. Lancet Infect Dis. 2010;10(2):83–92.

We have added this information (lines 114-115 in the clean revised manuscript): “As described in our generic protocol [8], we have included the selected cardiovascular disease codes most likely to be associated with RSV.”

• L124-128: The choice of J21.9 as part of the RSV proxy for children <2 is not supported by Japanese validation data. The RSV proportion for this population category has been shown to be around 60% and is highly variable (doi: 10.1371/journal.pone.0242302).

Response: The more generic bronchiolitis code (J21.9 or 466.1) is included in the RSV proxy to accommodate for the reduction in RSV testing during the peak and tail of the season, which we have observed in administrative databases in several countries. Additionally, it serves as an indicator marking the start of the RSV season (1-5). The proxies seek to accurately track the relative level of viral activity in the community, so the absolute value of the activity is less important than consistent measurement across the year to accurately track relative trends (Cf. generic protocol).

1) Calvo C, Pozo F, García-García ML, Sanchez M, Lopez-Valero M, Pérez-Breña P, et al. Detection of new respiratory viruses in hospitalized infants with bronchiolitis: a three-year prospective study. Acta Paediatr. 2010;99(6):883–887. doi: 10.1111/j.1651-2227.2010.01714.x.

2) Bandeira T, Carmo M, Lopes H, Gomes C, Martins M, Guzman C, et al. Burden and severity of children's hospitalizations by respiratory syncytial virus in Portugal, 2015–2018. Influenza Other Respir Viruses. 2023;17(1):e13066. doi: 10.1111/irv.13066.

3) Mansbach JM, Piedra PA, Teach SJ, Sullivan AF, Forgey T, Clark S, et al. Prospective multicenter study of viral etiology and hospital length of stay in children with severe bronchiolitis. Arch Pediatr Adolesc Med. 2012;166(8):700–706. doi: 10.1001/archpediatrics.2011.1669.

4) Walton NA, Poynton MR, Gesteland PH, Maloney C, Staes C, Facelli JC. Predicting the start week of respiratory syncytial virus outbreaks using real time weather variables. BMC Med Inform Decis Mak. 2010;10:68. doi: 10.1186/1472-6947-10-68.

5) Kenmoe S, Kengne-Nde C, Ebogo-Belobo JT, Mbaga DS, Fatawou Modiyinji A, Njouom R. Systematic review and meta-analysis of the prevalence of common respiratory viruses in children < 2 years with bronchiolitis in the pre-COVID-19 pandemic era. PLoS One. 2020;15(11):e0242302. doi: 10.1371/journal.pone.0242302.

The following sentence is slightly modified, adding the above references and information (lines 130-133 in the clean revised manuscript): “We included acute bronchiolitis, unspecified (J21.9), because RSV is responsible for the majority of bronchiolitis hospitalizations in this age group [23, 24] and was found to be a good indicator for the start of the RSV season [23-27].”

• L128-129: The influenza proxy is based on hospitalizations in adults +65 years, although the study population is +60 years.

Response: Thank you very much for flagging this. This was a typo, and we have replaced (line 134 in the clean revised manuscript) 65 years by 60 years.

• The manuscript does not describe the source, reliability, or testing practices underlying RSV and Influenza proxies.

Response: Please see lines 124-126 (in the clean revised manuscript): “As described in the generic protocol [8], we accounted for the circulation of selected pathogens (RSV and influenza) using hospital-based viral proxies for community-wide viral activity derived from MDV.” The underlying testing practices in each of the participated hospitals in MDV database are unknown. However, in general, hospital-based proxies have been found to be better predictors of RSV activity than surveillance data.

We have added a sentence, “The latter have been shown to better predict hospitalization than surveillance-based proxies [21].” (line 126 in the clean revised manuscript).

• L179-184: Excluding an entire year because data are incomplete could be suboptimal. A sensitivity analysis including partial 2019 data could strengthen the robustness of the results.

The 2015-2018 window (4 years) is relatively short.

Response: Please see lines 188-191 (in the clean revised manuscript) for the explanation that we have slightly modified by replacing “take into account” by “report”: “We modeled the hospitalization data until 30th June 2019 to make use of the most recent available data (prior to the COVID-19 pandemic). However, we did not report the results for the year 2019 since the data are incomplete….”

• L153-155: The scaling factor assumes DPC and non-DPC hospitals are comparable, which is unlikely (differences in patient age, specialization…).

• L.171-174: The manuscript does not discuss known structural differences between DPC and non-DPC hospitals (e.g., geriatric, rehabilitation facilities), which could significantly affect RSV burden estimates.

Response: Please note that the scaling factor described in lines 157-160 (in the clean revised manuscript) was used to project the estimated number of RSV-attributable hospitalizations to all DPC hospitals in Japan using a scaling factor corresponding to the number of all DPC hospitals in Japan divided by the number of DPC hospitals in the MDV database. We then carried out projections to all hospitals (including non-DPC hospitals) using two different types of scaling factor and acknowledging the uncertainty related to these projections and provided estimate ranges (please see lines 169-183, 258-272, 342-346 in the clean revised manuscript).

To clarify further, we modified the following sentences (lines 170-178 in the clean revised manuscript): “To project the IRs of RSV-attributable DPC hospitals to all hospitals, while accounting for the uncertainty linked to structural differences between DPC and non-DPC hospitals, we carried out two different projections by multiplying the estimated IRs of RSV-attributable DPC hospitals by different sets of year-specific scaling factors. The first scaling factor is defined as the total number of beds in all hospitals divided by the total number of beds in the DPC hospitals [25]. The second scaling factor was calculated by comparing the outcome-specific hospitalizations obtained from the MDV database to those from the JMDC Inc. payer-based database, which covers all hospital types in Japan and is representative of working adults younger than 65 years [27].”

• L168-170: The second scaling method, using adults aged 50-59 years, relies on an unproven assumption that patterns in 50-59 years reflect those +60. This needs justification.

Response: Thank you for raising this important point. According to our feasibility analysis, there is no single hospitalization data source available for Japan which accurately represents all adult age groups and all types of hospitals (DPC and non-DPC hospitals). The JMDC payer-based database is representative of all hospitalizations among working adults younger than 65 years of age, but contains limite

---

## [Decision Letter · Decision Letter 2]

19 Jan 2026

Dear Dr. Liang,

Thank you for submitting your manuscript to PLOS ONE. After careful consideration, we feel that it has merit but does not fully meet PLOS ONE’s publication criteria as it currently stands. Therefore, we invite you to submit a revised version of the manuscript that addresses the points raised during the review process.

We look forward to receiving your revised manuscript.

Kind regards,

Liling Chaw

Academic Editor

PLOS One

Journal Requirements:

Reviewers' comments:

Reviewer's Responses to Questions

**Comments to the Author**

Reviewer #3: (No Response)

2. Is the manuscript technically sound, and do the data support the conclusions?

Reviewer #3: Yes

3. Has the statistical analysis been performed appropriately and rigorously?

Reviewer #3: Yes

4. Have the authors made all data underlying the findings in their manuscript fully available?

Reviewer #3: No

5. Is the manuscript presented in an intelligible fashion and written in standard English?

Reviewer #3: Yes

Reviewer #3: L344: The statement regarding the “nationally representative of DPC” listed as a study strength should be moderated. The analysis relies exclusively on MDV data, which represent only a subset of DPC hospitals. Moreover, the DPC system itself covers approximately 54% of hospital beds in Japan. “nationally representative” appear overstated and should be avoided or more carefully qualified.

L204: The information concerning the total number of hospitalizations in Japan in 2017 does not appear essential (too broad) and could be omitted. The representativeness of the MDV data in relation to DPC hospitals and the national context is now sufficiently clarified in the Methods section.

L345-346: Presenting the six cardiovascular disease codes selected for this study as a strength should be moderated. Several comparable studies have included the full range of cardiovascular disease codes (I00–I99) (e.g., DOI: 10.1007/s40121-024-00951-0)

**Do you want your identity to be public for this peer review?** For information about this choice, including consent withdrawal, please see our Privacy Policy

Reviewer #3: No

---

## [Author Response · Author response to Decision Letter 3]

16 Feb 2026

Dr. Liling Chaw

Academic Editor

PLOS ONE

Date: 16 February 2026

Dear Dr. Chaw,

We are grateful for the opportunity to revise the manuscript in light of the comments from Reviewer #3. We have responded to all comments as itemized in our point-by-point reply below in blue. All changes are saved in tracked changes in the revised version (file name with: Revised_Manuscript_with_Track_Changes) of our manuscript. An unmarked version of our revised manuscript without tracked changes (file name with: Manuscript) is also shared.

Sincerely yours,

Caihua Liang

on behalf of the authors

Reviewers' comments:

Reviewer's Responses to Questions

Comments to the Author

1. If the authors have adequately addressed your comments raised in a previous round of review and you feel that this manuscript is now acceptable for publication, you may indicate that here to bypass the “Comments to the Author” section, enter your conflict of interest statement in the “Confidential to Editor” section, and submit your "Accept" recommendation.

Reviewer #3: (No Response)

Response: We thank Reviewer #3 for the helpful feedback. We have addressed the additional concerns raised by Reviewer #3; please find our detailed answers in section 6.

2. Is the manuscript technically sound, and do the data support the conclusions?

Reviewer #3: Yes

Response: We thank Reviewer #3 for the confirmation.

3. Has the statistical analysis been performed appropriately and rigorously?

Reviewer #3: Yes

Response: We thank Reviewer #3 for the confirmation.

4. Have the authors made all data underlying the findings in their manuscript fully available?

Reviewer #3: No

Response: The Data Availability Statement has already been updated to explain why the datasets are not publicly available and how they can be requested. We share the Data Availability Statement, again below:

Data Availability Statement: “The datasets generated during and/or analyzed during the current study are not publicly available. The data are owned by a third-party, Medical Data Vision (MDV) Co., Ltd, and the authors do not have permission to share the data. The data are available upon request to MDV (https://en.mdv.co.jp; contacts: Hiroyuki Tada [email: tada@mdv.co.jp] and Yuki Santo [email: santo_yuki@mdv.co.jp]).

The statistical codes are available upon request to the following contacts: statistics department, P95 (email: stat@p-95.com), and Hannah Volkman (Pfizer Inc; email: Hannah.Volkman@pfizer.com).”

5. Is the manuscript presented in an intelligible fashion and written in standard English?

Reviewer #3: Yes

Response: We thank Reviewer #3 for the confirmation.

6. Review Comments to the Author

Reviewer #3:

• L344: The statement regarding the “nationally representative of DPC” listed as a study strength should be moderated. The analysis relies exclusively on MDV data, which represent only a subset of DPC hospitals. Moreover, the DPC system itself covers approximately 54% of hospital beds in Japan. “nationally representative” appear overstated and should be avoided or more carefully qualified.

Response: We thank Reviewer #3 for the comment. We have adapted the sentence to take this into account. Please see L342-344 (in the revised clean manuscript): 'We used a database that includes DPC hospitalizations across different regions throughout Japan, which reduced the possibility of sampling bias.’

• L204: The information concerning the total number of hospitalizations in Japan in 2017 does not appear essential (too broad) and could be omitted. The representativeness of the MDV data in relation to DPC hospitals and the national context is now sufficiently clarified in the Methods section.

Response: The sentence L204 has been deleted.

• L345-346: Presenting the six cardiovascular disease codes selected for this study as a strength should be moderated. Several comparable studies have included the full range of cardiovascular disease codes (I00–I99) (e.g., DOI: 10.1007/s40121-024-00951-0)

Response: We thank Reviewer #3 for the comment. We have adapted the sentence to take this into account. Please see L344-346 (in the revised clean manuscript): ‘We also used an extensive list of ICD-10 codes, corresponding to broad definitions for respiratory outcomes and more specific for cardiovascular diseases, thereby improving the sensitivity of detecting RSV-related outcomes [16, 43-45].’

To support this amendment, we have added new references: nos. 43-45.

7. PLOS authors have the option to publish the peer review history of their article (what does this mean?). If published, this will include your full peer review and any attached files.

Response: We are fine to publish the peer review history of our article.

Do you want your identity to be public for this peer review? For information about this choice, including consent withdrawal, please see our Privacy Policy.

Reviewer #3: No

---

## [Decision Letter · Decision Letter 3]

19 Feb 2026

Estimation of Respiratory Syncytial Virus-attributable hospitalizations among older adults in Japan between 2015 and 2018: an administrative health claims database analysis

PONE-D-25-13816R3

Dear Dr. Liang,

We’re pleased to inform you that your manuscript has been judged scientifically suitable for publication and will be formally accepted for publication once it meets all outstanding technical requirements.

Kind regards,

Liling Chaw

Academic Editor

PLOS One

Additional Editor Comments (optional):

Reviewers' comments:

Reviewer's Responses to Questions

**Comments to the Author**

Reviewer #3: All comments have been addressed

2. Is the manuscript technically sound, and do the data support the conclusions?

Reviewer #3: Yes

3. Has the statistical analysis been performed appropriately and rigorously?

Reviewer #3: Yes

4. Have the authors made all data underlying the findings in their manuscript fully available?

Reviewer #3: No

5. Is the manuscript presented in an intelligible fashion and written in standard English?

Reviewer #3: Yes

Reviewer #3: (No Response)

**Do you want your identity to be public for this peer review?** For information about this choice, including consent withdrawal, please see our Privacy Policy

Reviewer #3: No

---

## [Editor Report · Acceptance letter]

PONE-D-25-13816R3

PLOS One

Dear Dr. Liang,

I'm pleased to inform you that your manuscript has been deemed suitable for publication in PLOS One. Congratulations! Your manuscript is now being handed over to our production team.

Kind regards,

on behalf of

Dr. Liling Chaw

Academic Editor

PLOS One